# *Ruta graveolens*: Boost Melanogenic Effects and Protection against Oxidative Damage in Melanocytes

**DOI:** 10.3390/antiox12081580

**Published:** 2023-08-08

**Authors:** Pazilaiti Ainiwaer, Zuopeng Li, Deng Zang, Lan Jiang, Guoan Zou, Haji Akber Aisa

**Affiliations:** 1State Key Laboratory Basis of Xinjiang Indigenous Medicinal Plants Resource Utilization, Key Laboratory of Plants Resources and Chemistry of Arid Zone, Xinjiang Technical Institute of Physics and Chemistry, Chinese Academy of Sciences, South Beijing Road 40-1, Urumqi 830011, China; pazilaiti19@mails.ucas.ac.cn (P.A.); lizp@ms.xjb.ac.cn (Z.L.); zangdeng@ms.xjb.ac.cn (D.Z.); jianglan@ms.xjb.ac.cn (L.J.); gazou@ms.xjb.ac.cn (G.Z.); 2University of Chinese Academy of Sciences, No.19(A) Yuquan Road, Beijing 100049, China

**Keywords:** *Ruta graveolens*, chemical constituents, vitiligo, melanogenesis, ER stress

## Abstract

Vitiligo, an acquired depigmentation disorder, is characterized by the loss of functional melanocytes and epidermal melanin. In recent years, research has focused on promoting melanin biosynthesis and protecting melanocytes to reduce stress-related damage for the purpose of applying it to vitiligo treatment. *Ruta graveolens* L. has been utilized as a medicinal herb in diverse traditional medicine systems to address conditions like vitiligo. In this investigation, we isolated and purified 16 unique alkaloid compounds from the chloroform extracts of *R. graveolens*, encompassing a new quinoline alkaloid and several recognized compounds. Bioactivity analysis showed that compound **13**, an alkaloid derived from *R. graveolens*, promotes melanin production while protecting PIG3V melanocytes against 4-tert-butylphenol (4-TBP)-induced oxidative damage by downregulating endoplasmic reticulum (ER) stress and pro-inflammatory cytokines through interleukin-6 (IL-6) regulation. Additionally, the compound suppressed the expression of Bip, IRE1, p-IRE1, and XBP-1 proteins, suggesting a potential antioxidant function. These findings suggest that compound **13** isolated from *R. graveolens* can augment melanogenesis in melanocytes, reduce endoplasmic reticulum (ER) stress, and ameliorate vitiligo exacerbation. The melanogenic activity observed in the chloroform fraction emphasizes *R. graveolens*’s potential as a novel therapeutic target for vitiligo treatment, warranting further exploration in future studies.

## 1. Introduction

Vitiligo is a common depigmenting disorder, manifested as localized skin or mucosal white patches stemming from the loss of functional melanocytes [1]. Vitiligo’s pathogenesis is multifaceted, and the most widely accepted theory posits that autoimmune responses target melanocytes based on genetic susceptibility to melanocyte damage coupled with antigen exposure resulting from abnormal oxidative stress responses [2]. Currently, clinical treatments involve drug therapy, phototherapy, surgical transplantation, and other methods. While these treatments may temporarily restore pigmentation in affected areas, no definitive cure has been established thus far.

Melanocytes, located in the deepest basal cell layer of the epidermis, produce and release melanin [3]. Essential for epidermal homeostasis, melanin protects epidermal cells from ultraviolet radiation-induced damage [4]. However, vitiligo is characterized by the appearance of depigmented skin patches [5]. Numerous studies link oxidative stress to vitiligo onset. Some suggest that misfolded protein accumulation in the endoplasmic reticulum (ER) activates the unfolded protein response (UPR), causing oxidative stress implicated in vitiligo pathogenesis [6]. While UPR primarily restores cellular homeostasis for normal cell survival and differentiation, chronic ER stress-induced UPR can adversely affect cells, activating apoptosis [7]. Melanocytes of vitiligo patients exhibit morphological and biological abnormalities, such as dilated ER profiles and increased sensitivity to oxidative stress exposure compared to control cells [8]. PIG3V, as an immortalized human vitiligo melanocyte characterized by ER dilation and intrinsic defects, is more susceptible to oxidative stress [9]. Sustained melanocyte stress will lead to melanocyte apoptosis while also initiating the targeting of melanocytes by the immune system [7]. 4-Tertiary butyl-phenol (4-TBP), a vitiligo trigger, induces reactive oxygen species (ROS) generation, disrupting melanocyte ER homeostasis and accumulating protein folding perturbations that activate UPR [10]. This process involves inositol-requiring enzyme 1 (IRE1), PKR-like ER kinase (PERK), RNA-dependent protein kinase-like ER eukaryotic initiation factor-2a (eIF2a), and activating transcription factor (ATF)-6 [11]. Although 4-TBP initially promotes ER homeostasis restoration and cell survival, sustained stress turns it into a cell death mechanism [12]. IRE1, a highly conserved UPR signaling node, activates in the presence of UPR, undergoing sequential dimerization, trans-autophosphorylation, and endoribonuclease activation [13]. Subsequently, IRE1 endoribonucleases splice X-box-binding protein 1 (XBP-1), forming a primary substrate [14]. After melanocyte exposure to chemical triggers, XBP-1 expression increases, elevating immune mediator interleukin-6 (IL-6) production. Increased IL-6 levels may contribute to vitiligo’s autoimmune-mediated progression [15]. Consequently, recent studies emphasize reducing protein folding machinery load in ER stress and UPR as potential therapeutic targets for vitiligo treatment [11].

*Ruta graveolens* L. is a shrubby perennial plant in the Rutaceae family. Native to the Mediterranean region, it is now cultivated as a medicinal and ornamental plant in Asia, South America, and Europe [11]. In Unani medicine, *R. graveolens* is traditionally reported to possess abortifacient, anti-vitiligo, and blood circulation-enhancing properties when applied topically, as well as anti-inflammatory effects and relief for joint and gout pain [16]. In Ayurvedic medicine, it is used as an antispasmodic and digestive agent [17]. Traditional Chinese medicine prescribes *R. graveolens* to treat diseases such as psoriasis, vitiligo, and lymphoma [18]. Due to its long history of medicinal use, *R. graveolens* has attracted significant research interest. Recent pharmacological activity studies reveal that *R. graveolens* exhibits antioxidant, anti-inflammatory [19], antidiabetic [20], antibacterial [21], antiandrogenic [22], and insecticidal effects [23].

The primary chemical constituents of *R. graveolens* are coumarins and alkaloids [24]. Psoralen, a coumarin component, has been clinically utilized for vitiligo treatment. Nonetheless, studies on vitiligo treatment involving alkaloid constituents are scarce. Consequently, this study aims to expand the understanding of *R. graveolens*’ pharmacodynamic basis for vitiligo treatment by isolating its alkaloid components and investigating their anti-vitiligo activity through screening and mechanical analysis. Our objectives include isolating alkaloids compounds from *R. graveolens*, evaluating their melanogenic effects on B16 melanocytes in vitro, and examining the most active compound’s impact on the unfolded protein response (UPR) signal transduction pathway in human melanocytes to reveal its mechanism of action anti-vitiligo.

## 2. Materials and Methods

### 2.1. Chemicals

DMEM high glucose medium, fetal bovine serum (FBS), and the penicillin–streptomycin solution were all obtained from Gibco (Life Technologies, Grand Island, NY, USA). 4-Nitrophenyl phosphate, disodium salt (PNPP), Dimethyl sulfoxide (DMSO), and other reagents were purchased from Sigma (St. Louis, MO, USA). Antibodies include the following: anti-EIF2*α* (S51), (D9G8), anti-Bip (C50B12), anti-IRE1 (14C10), anti-Spliced-XBP-1 (D2C1F), anti-ATF-4 (D4B8), anti-ATF-6 (AZ8V), anti-PERK (D11A8), were purchased from Cell Signaling Technology (Danvers, MA, USA). anti-p-IRE1 (ser724, Novus, Colorado, USA), and *β*-actin (#BM3873, BOSTER Biological Technology, Wuhan, China). Inhibitors include the following: salicylaldehyde (S5391-100, Solarbio, Beijing, China) and rapamycin (abs47000712, Absin Bioscience Inc. Shanghai, China).

### 2.2. Plant Material

The aerial parts of *R. graveolens* (Leaves, stems, and flowers) were gathered in Qapqal Xibe Autonomous County, Xinjiang Province, China, at the beginning of June 2018. The plant was determined by Dr. Lu Chunfang. A voucher specimen (No. WY02260) was stored in the herbarium of Xinjiang Technical Institute of Physics and Chemistry, Chinese Academy of Sciences (CTIPC).

### 2.3. Equipment

Spectra data were obtained by using a Varian MR-400 and VNMRS-600 instrument (Varian, Palo Alto, CA, USA). Tetramethylsilane (TMS, *δ* = 0 ppm) was the internal standard of the chemical shift. Silica gel column chromatography (Qingdao Haiyang Chemical Co.), a preparative high-performance liquid chromatography (HPLC), was conducted using a Shimadzu type LC-20AT (Shimadzu, Tokyo, Japan) with an Octadecylsilyl (ODS) column chromatography (C18 reversed-phase silica gel, SP-120-50-ODS-B, Daisogel, Japan), which were were used to purify the compounds.

### 2.4. Extraction and Isolation

The aerial parts of *R. graveolens* (20.0 kg) were extracted with EtOH (100 L) 3 times. After evaporating the ethanol, the resulting extract was distributed in the water and defatted by petroleum ether, followed by successively extracted with chloroform, ethyl acetate, and n-butanol. Then the corre-sponding layers were acquired after concentration in a vacuum.

The CHCl_3_ fraction (303.9 g) was separated by CH_2_Cl_2_/CH_3_OH (100:1–0:100, *v*/*v*) to provide 13 fractions (Fr. 1~Fr. 13) on the silica gel column. Fr. 10 was separated by normal phase CC with CH_2_Cl_2_/CH_3_OH (80:1–1:1, *v*/*v*) to yield 3 subfractions (Fr. 10–1~Fr. 10–3). Thereafter, compounds **1** (3.5 mg), **2** (6.1 mg), **5** (7.8 mg), **6** (7.0 mg), **7** (4.8 mg), and **14** (3.3 mg) were obtained from Fr. 10–1 using p-HPLC. Compounds **3** (3.2 mg), **4** (5.3 mg), **6** (10.1 mg), **13** (13.4 mg), and **16** (2.4 mg) were obtained from Fr. 10–2 using p-HPLC. Fr. 6 was subjected to silica gel CC to provide 3 sub-fractions (Fr. 6–1~Fr. 6–3). Fr. 6–2 was separated by preparative thin-layer chromatography (CH_2_Cl_2_/CH_3_OH, 8:1) to obtain **15** (2.8 mg). Fraction of Fr. 6–3 was separated with CH_2_Cl_2_/CH_3_OH (40:1, *v*/*v*) on a silica gel CC and then purified by p-TLC (CH_2_Cl_2_/CH_3_OH, 6:1) to yield **9** (3.5 mg) and **8** (1.8 mg).

Fr. 7 was separated by normal phase CC with CH_2_Cl_2_/CH_3_OH (60:1–0:1, *v*/*v*) to yield 4 subfractions (Fr. 7–1~Fr. 7–4). Compounds **10** (3.6 mg), **11** (2.6 mg), and **12** (17.7 mg) were obtained from Fr. 7–1 using HPLC.

### 2.5. Cell Treatments and Culture

The murine melanoma B16 cell line (Cat#TCM2) was cultured Dulbecco’s Modified Eagle Medium (DMEM; Gibco Life Technologies, Waltham, MA, USA) supplemented with 10% heat-inactivated fetal bovine serum (FBS; Gibco Life Technologies), 100 units/mL penicillin G, and 100 µg/mL streptomycin (Gibco BRL, Grand Island, NY, USA) at 37 °C in a 5% CO_2_ incubator (Thermo Fisher Scientific, Waltham, MA, USA). PIG3V melanocytes were kindly provided by Dr. Caroline Le Poole (Loyola University Chicago, Maywood, IL, USA) and maintained in Medium 254 with human melanocyte growth supplement (Gibco Life Technologies), 5% FBS, 100 U/mL penicillin G, and 100 µg/mL streptomycin at 37 °C in a 5% CO_2_ environment.

### 2.6. Cell Viability Assay

The viability of B16 cells was assessed using a CCK-8 assay, as previously described [25]. B16 cells were seeded at a density of 5 × 10^3^ cells per well in 96-well plates and incubated at 37 °C at concentrations ranging from 1 to 50 µM for 24 h. Subsequently, the culture medium was replaced with CCK-8 solution (10 µL, Absin, Shanghai, China), and the cells were incubated for an additional 2 h at 37 °C. Absorbance was measured at 450 nm using a Spectra Max M5 microplate reader (Molecular Devices Company, San Diego, CA, USA). An equal volume of untreated cells served as a blank control. All experiments were performed in triplicate.

### 2.7. Assay for Relative Intracellular Melanin Concentration and Tyrosinase Activity

We conducted the in vitro melanin content assay experiment in this study using previously described methods [26]. In brief, B16 cells were detached from the cultured flask using trypsin-EDTA solution, counted with a hemocytometer, and centrifuged to obtain a pellet, which was dissolved in 1 mM NaOH. Melanin content was determined by measuring optical density at 405 nm. Tyrosinase activity was assessed by evaluating the rate of L-DOPA oxidation, as described by [27], with slight modifications. To determine intracellular tyrosinase activity, B16 melanoma cells were seeded in 6-well plates at a density of 1 × 10^5^ cells/well and cultured for 24 h. The cells were then incubated with the samples (polarity fractions extract and compounds) for 48 h. Following treatment, the cells were lysed with 1% Triton X-100/PBS, frozen at −20 °C for 20 min, thawed, and mixed. Cell lysates were combined with 10 μL 0.1% L-DOPA in a 96-well plate, incubated at 37 °C for 20 min, and absorbance was measured at 490 nm to monitor dopachrome formation. 8-Methoxypsoralen’s (8-MOP) was the positive control, and Dimethyl sulfoxide (DMSO) was the benchmark solvent used for preparing stock solutions of compounds intended for assessing melanin concentration and tyrosinase activity.

### 2.8. Western Blotting

B16 cells (2 × 10^5^ cells/well) were seeded in 6-well plates, incubated for 24 h, and treated with various concentrations of component **13** (1, 10, and 50 µM) for 48 h. Whole-cell protein extracts were prepared using RIPA buffer, and a BCA assay kit was used to measure protein concentration. Proteins were separated on 10% SDS-PAGE gels, transferred onto a polyvinylidene difluoride (PVDF) membrane, and immunoblotted with the specific primary antibodies overnight at 4 °C, followed by incubation with secondary antibodies for 1 h. The primary antibodies used were as follows: anti-EIF2*α* (S51), (D9G8), anti-Bip (C50B12), anti-IRE1 (14C10), anti-p-IRE1 (ser724), anti-Spliced-XBP-1 (D2C1F), anti-ATF-4 (D4B8), anti-ATF-6 (AZ8V), anti-PERK (D11A8), and *β*-actin (#BM3873). Protein bands were visualized using ECL reagents. Each experiment was run three times. Protein expression levels were semi-quantified and photographed using the ChemiDoc MP Imaging System (Bio-Rad Laboratories Inc., Hercules, CA, USA).

### 2.9. Cytotoxicity and Oxidative Injury Assessment in 4-TBP-Induced PIG3V Melanocytes

PIG3V melanocytes were seeded in 96-well plates (5 × 10^3^ cells/well) and cultured in Medium 254 with 10% FBS and 1% HMGS-2 for 24 h. The cells were treated with component **13** at concentrations of 0, 1, 5, 10, 50, and 100 μM for 12 h. Subsequently, the cells were co-treated with 300 μM of 4-TBP to induce oxidative stress. The CCK-8 assay was used to measure cell viability.

### 2.10. ROS Assessment in 4-TBP-Induced PIG3V Melanocytes

In order to generate ROS in PIG3V cells, a DCFH-DA probe (Nanjing Jian Cheng Biotechnology Co., Ltd., E004, Nanjing, China) was utilized PIG3V cells were seeded in 6-well plates and cultured in Medium 254 with 10% FBS and 1% HMGS-2 for 24 h. The cells were treated with a 10 µM component **13** and 300 μM 4-TBP. Subsequently, the culture medium was removed, and the cells were washed twice with PBS before incubation with a 10 μM DCFH-DA solution in serum-free medium for 30 min at 37 °C. The cells were washed three times to thoroughly remove excess DHE-DA. Micrographs were captured using a fluorescence microscope (Olympus, Tokyo, Japan) with excitation and emission wavelength of 500 nm and 530 nm.

### 2.11. Cytokine IL-6 Secretion Assessment in 4-TBP-Induced PIG3V Melanocytes

PIG3V cells were treated with component **13** at various concentrations (1, 5, and 10 μM) and with 4-TBP at a concentration of 300 μM, as described previously. The treatment duration was specified according to the experimental design. The cell culture medium was subsequently collected to analyze IL-6 levels using ELISA kits provided by R&D, Noves, CA, USA, following the manufacturer’s instructions.

### 2.12. Evaluation of UPR Modulators in 4-TBP-Induced PIG3V Melanocytes

PIG3V melanocytes were seeded in 6-well plates at a density of 4×10^5^/well and cultured and treated as described previously. Protein component **13** were isolated and analyzed using Western blot with the following primary antibodies: anti-EIF2*α* (S51), (D9G8), anti-Bip (C50B12), anti-IRE1 (14C10), anti-p-IRE1 (ser724), anti-Spliced-XBP-1 (D2C1F), anti-ATF-4 (D4B8), anti-ATF-6 (AZ8V), anti-PERK (D11A8), and *β*-actin (#BM3873). PIG3V melanocytes were exposed to 80 μM of SA (Absin, China, abs47000712) or 100 nM of rapamycin (Solarbio, China, S5391-100) 2 h before exposure to 4-TBP and with or without the component **13**. The cell culture medium was subsequently collected to analyze IL-6 levels using ELISA kits provided by R&D, Noves, CA, USA, following the manufacturer’s instructions.

### 2.13. Statistical Analysis

All data were expressed as mean ± standard error of the median (SEM), and statistical analysis was performed using one-way ANOVA and Tukey’s multiple comparison tests with GraphPad Prism 9 (La Jolla, CA, USA). The *p*-values established significant differences at *p* < 0.05.

## 3. Results

### 3.1. Identification of the Components

By using various chromatographic methods, particularly the preparative high-performance liquid chromatography (p-HPLC), sixteen chemical constituents, including a new quinoline alkaloid, were successfully isolated from the chloroform layer of *R. graveolens.* The red numeration signifies the new compound (Figure 1).

Compound **1** was obtained as a colorless powder: IR: *v*_max_ = 3268, 2923, 1647, 1602, 1456, 1260, 1088, 1030, 872 cm^−1^; UV (MeOH) *λ*max (log *ε*): 229 (2.80); [*α*]20 D +12.0 (*c* 0.05, MeOH). According to the ^13^C-NMR and ^1^H-NMR data (Table 1) and the HR-ESI-MS ([M + H] ^+^
*m*/*z* 276.1225, calcd. as 276.1230), the formula of 1 was decided as C_15_H_17_NO_4_.

The ^1^H-NMR (Table 1) of compound **1** displayed aromatic proton at *δ*_H_ 7.19 (1H, dd, *J* = 9.0, 2.8 Hz, H-7), 7.56 (1H, d, *J* = 9.0 Hz, H-8), and 7.23 (1H, d, *J* = 2.8 Hz, H-5), together with a methoxy proton signal at *δ*_H_ 4.04 (3H, s, 4-OCH_3_), a methylene proton signal at *δ*_H_ 3.90 (1H, dd, *J* = 6.6, 5.1 Hz, H-3′), and methine proton signals at *δ*_H_ 3.21 (1H, dd, *J* = 17.1, 5.1 Hz, H-4′*β*) and 2.94 (1H, dd, *J* = 17.1, 6.6 Hz, H-4′*α*). Two methyl proton signals emerge at *δ*_H_ 1.41 (3H, s, 2′*α*-CH_3_) and 1.44 (3H, s, 2′*β*-CH_3_). ^13^C-NMR and HMQC analyses revealed the presence of 15 carbon atoms consisting of the following functional groups: three aromatic carbons (*δ*_C_ 128.5, 122.9, 104.4), four aromatic carbons bonded to oxygen or nitrogen (*δ*_C_ 164.0, 159.6, 155.3, 142.5), three quaternary carbons (*δ*_C_ 122.7, 109.5, 80.4), one methylene resonance (*δ*_C_ 69.2), one methine resonance (*δ*_C_ 27.5), two methyl resonances (*δ*_C_ 26.1, 22.2), and one methoxy resonance (*δ*_C_ 61.6).

In the ^1^H-^1^H COSY spectrum, H-7 (*δ*_H_ 7.19) correlates with H-8 (*δ*_H_ 7.56), and the splitting of H-7 into a dd peak and H-8 into a d peak suggests the presence of a 1,2,4-trisubstituted benzene ring. H-3′ correlates with H-4′, suggesting that the methylene and hypomethyl groups are linked. In the HMBC spectrum Figure 2, H-7 correlates with C-5/C-8a, H-8 correlates with C-6/C-4a, and H-5 (*δ*_H_ 7.23) correlates with C-7/C-8a/C-4, further suggesting that the compound has a 1,2,4-trisubstituted benzene ring with a six-membered azepine ring attached to the 1,2 positions of the benzene ring and the 4 position of the benzene ring substituted with a hydroxyl group. Methoxy (*δ*_H_ 4.01) correlates with C-4 (*δ*_C_ 164.0), suggesting that methoxy is located at the C-4 position of the quinoline alkaloids; H-4′ (*δ*_H_ 3.21 and 2.94) is associated with C-3′/C-2′/C-2/C-3/C-4, indicating the presence of the pyran ring and the connection of the quinoline ring via the 2,3 position; and 2′-CH_3_ (*δ*_H_ 1.41 and 1.44) is associated with C-2′/C-3′/C-3′correlation, suggesting the presence of two methyl groups attached to the C-2′ position and a hydroxyl group attached to the C-3′ position. The above HMBC correlation defines the planar structure of the compound Figure 1. An analysis of the literature [28] revealed that compound 1 is like the known compound 3,4-dihydro-3-hydroxy-5-methoxy-2,2-dimethyl-2H-Pyrano[2,3-*b*] quinoline, with the difference that compound 1 has an additional hydroxyl substitution at the C-6 position on the benzene ring.

The relative configuration of the compound was analyzed by NOESY correlation. In the NOESY spectrum, the correlation signal of H-3′/2′*β*-CH_3_/H-4′*β* suggests that the above hydrogens are located on the same side, designated as the *β*-position, while the other methyl group on C-2′ (*δ*_H_ 1.41) and the other hydrogen on C-4′ (*δ*_H_ 2.94) is then at the *α*-position, and the correlation signal of H-4′*α* (*δ*_H_ 2.94)/2′*α*-CH_3_ (*δ*_H_ 1.41) verifies the above inference. Due to the presence of a chiral carbon C-3′ in this structure, the compound was confirmed not to be a racemate by the high-performance liquid chromatography chiral analysis column. In order to determine the absolute configuration of compound **1**, the circular dichroism of the compound was determined. The theoretical ECD (Electronic Circular Dichroism) of 3′*S*-1 and 3′*R*-1 was then calculated by applying quantum chemical calculations (method set to TDDFT method, pbe0m4/def-TZVP) using the planar and minimum energy structures as a benchmark, and the measured experimental CD curves were compared with the theoretical. The ECD results were compared, and the stereo-configuration was determined based on the agreement between the measured and theoretical values, and the results showed that the calculated ECD curve for the stereo-configuration 3′*S*-1 agreed with the measured CD spectrum Figure 3. Therefore, the absolute configuration of the compound was determined to be 3′*S*, and the compound was identified as new by a SciFinder search. The systematic name of the compound is 5-methoxy-2,2-dimethyl-3,4-dihydro-2H-pyrano [2,3-*b*] quinoline-3,7-diol and is named rutagrarin A.

Sixteen alkaloids were simultaneously obtained from the chloroform fraction of the aerial parts of this herb, and the structures of these components were determined as new compound rutagrarin A (**1**) and known compounds arborinin (**2**) [29], 1,3-dihydroxy-2-methoxy-10-methyl-9-acridone (**3**) [30], 2-n-heptyl-4-hydroxyquinoline (**4**) [31], 2-heptyl-4(1H)quinolone (**5**) [32], 2-[4′-(3′,4′-methylenedioxyphenyl)butyl]-4-quinolone (**6**) [33], graveolin (**7**) [34], indole-3-carboxylic acid (**8**) [35], xylogranatinin (**9**) [36], gravacridondiol (**10**) [37], 6-methoxy-7-hydroxydictamnine (**11**) [38], skimmianine (**12**) [39], kokusaginine (**13**) [40], ribaline (**14**) [41], ribalinidina (**15**) [42], and gravacridondiol-18-*β*-*D*-glucoside (**16**) [43]. Among these, isolated compounds **2**, **3**, **8,** and **10** were obtained from this herb for the first time.

### 3.2. Melanogenic Effects of R. graveolens Fractions in B16 Cells

We evaluated the cytotoxicity of different polarity fractions (petroleum ether, chloroform, ethyl acetate, and *n*-butanol) at a concentration of 50 μg/mL. Subsequently, we assessed their activity at the same concentration. Our results demonstrated that at a concentration of 50 μg/mL, the cell survival rate for B16 cells was greater than 80%, with no significant difference compared to the blank control group. This indicates that the extracts exhibit no cytotoxic effects at this concentration, thus deeming it a safe concentration. Further analysis of melanin content revealed that the chloroform extract (YX-L) and the petroleum ether extract (YX-S) promoted melanin production at 50 μg/mL, reaching 195.1% ± 5.120 and 178.7% ± 6.239, respectively. These rates surpass the positive control 8-Methoxypsoralen’s (8-MOP) activation rate of 134.8% ± 1.492 at 50 μM (Figure 4).

### 3.3. Effects on Relative Melanin Content and Tyrosinase Activity of R. graveolens Fractions in B16 Cells

To determine the most suitable concentration range for promoting melanin production, we tested various concentrations (1, 10, 50 μg/mL) of chloroform and petroleum ether fractions. We found that both the petroleum ether and chloroform extracts could consistently enhance melanin content and tyrosinase activity, exhibiting a strong concentration dependence (Figure 5). The chloroform fraction demonstrated a significant increase in melanin content and tyrosinase activity at the concentration of 10–50 µg/mL. At 10 µg/mL, the activation rates for tyrosinase and melanin content were 136.1% and 129.0%, respectively. At 50 µg/mL, these rates increased to 195.1% and 192.7%, respectively. Similarly, the petroleum ether extract showed notable promotion of melanin content and tyrosinase activity. At 10 µg/mL, the activation rates for tyrosinase and melanin content were 124.8% and 124.6%, respectively. At 50 µg/mL, these rates increased to 178.7% and 168.3%, respectively.

Our activity screening outcomes revealed that fractions of varying polarity enhanced melanogenesis and tyrosinase activity in a concentration-dependent manner in B16 cells. Upon reaching a concentration of 50 μg/mL, no cytotoxic effects were observed, leading us to select a concentration range of 10–50 μg/mL of the polarity fractions for subsequent experimental validation.

The screening results for active components in *R. graveolens* fractions reveal that the chloroform and petroleum ether extracts have a significant impact on promoting melanin production compared to other extracts. This implies that key active components, potentially driving *R. graveolens*’ effects, could be contained in the chloroform and petroleum ether extracts. Consequently, our focus is on determining the active ingredients within the chloroform extract and exploring its potential mechanisms for treating vitiligo.

### 3.4. Melanogenic Effects of Compounds ***1–16*** in B16 Cells

Compounds **1**–**16** were tested for their ability to activate melanin production and affect tyrosinase activity. The results, shown in Figure 6, demonstrate varying levels of melanin promotion at a concentration of 50 μM. Among the compounds, kokusaginine (**13**) exhibited the highest activation rates for melanin content and tyrosinase activity, reaching 185.3%, which showed a significant difference (*p* < 0.0001) compared to the blank control rate of 148.9%. Therefore, we further investigated the effects of different concentrations of compound **13** on the B16 cell survival rate, focusing on the concentration range that resulted in the highest survival rate for B16 cells, as well as its influence on melanin content within these cells.

### 3.5. Cell Toxicity of Compound ***13***

We utilized CCK-8 assays to assess the effects of different concentrations (0–200 µM) of compound **13** on B16 cell viability. The results, depicted in Figure 7, indicate that within the concentration range of 1–50 µM, compound **13** has no significant impact on B16 cell viability after 24 h. The survival rate of the experimental group’s cells remained greater than 80%, with no statistical difference compared to the blank group (*p* > 0.05). This demonstrates that compound **13** does not significantly affect the viability of B16 cells within the concentration range of 1–50 µM. However, at a concentration of 200 µM, the survival rate of B16 cells treated with compound **13** drops below 80%, showing a significant difference compared to the blank group (**** *p* < 0.0001). Based on these cell viability results, we selected 1, 10, and 50 µM as safe concentrations to conduct subsequent activity and mechanism of action experiments.

### 3.6. Effect on Melanin Content and Tyrosinase Activity of ***13***

We employed the L-DOPA method and NaOH cracking method to determine the effects of varying concentrations of compound **13** on melanin content and tyrosinase activity in B16 cells. The results are presented in Figure 8. Treatment of B16 cells with different concentrations of compound **13** resulted in a significant increase in relative melanin content. At a concentration of 50 µM, compound **13** enhanced melanin content in B16 cells to 170.1% ± 8.493% (**** *p* < 0.001), which is notably higher than the positive control group 8-MOP. The difference in melanin production rates between compound **13** and 8-MOP is statistically significant (*** *p* < 0.001). However, compound **13** did not exhibit any association with tyrosinase activity. We hypothesize that compound **13** may not stimulate tyrosinase to activate melanin synthesis.

Compound **13** is a quinoline alkaloid, which has been reported to possess antioxidant effects [44]. We can be sure that only promoting melanogenesis is not the unique target in vitiligo therapy. It urges us to seek another mechanism of our compounds for the treatment of vitiligo. Therefore, we further explored whether the compounds have an effect on oxidative injury.

### 3.7. Protective Effects of Compound ***13*** on 4-TBP-Induced Oxidative Injury in PIG3V Melanocytes

In this study, we employed 4-TBP, a known inducer of vitiligo, to trigger ER stress and activate the UPR in human melanocytes, leading to an increase in cytokine IL-6 expression. We then evaluated whether compound **13** could decrease the load on the folding machinery in ER stress and UPR. Our results indicated that 4-TBP at a concentration of 300 μM resulted in 80% cell viability in PIG3V cells after 24 h. In addition, compound **13** at concentrations of 5 and 10 μM increased the cell survival rate in PIG3V cells exposed to 4-TBP (Figure 9); compound **13** at concentrations up to 100 µM did not affect cytotoxicity in PIG3V cells. These results indicated that compound **13** protected PIG3V melanocytes against 4-TBP-induced oxidative injury. Therefore, compound **13** was selected for an in-depth analysis of 4-TBP-induced oxidative injury.

### 3.8. Compound ***13*** Inhibited ROS Generation and Cytokine Expression Induced by 4-TBP in PIG3V Melanocytes

According to a previously reported method [10], 4-TBP at a concentration of 300 μM was selected to treat melanocytes. ROS generation in melanocytes may be a provocative mechanism responsible for cytotoxicity induced by 4-TBP. In our study, the incubation of PIG3V melanocytes with 300 μM of 4-TBP for 6 h resulted in a significant increase in ROS generation. Moreover, the co-treatment of 4-TBP-treated PIG3V melanocytes with compound **13** resulted in decreased ROS production (Figure 10C,D). After melanocytes were treated with 4-TBP, cytokine expression levels, especially IL-6 expression, were significantly upregulated at 6 h. In order to verify the effects of compound **13** on the expression of IL-6, we performed ELISA. Following the treatment with 4-TBP at a concentration of 300 μM, IL-6 expression exhibited significant upregulation at the 6 h mark. However, when cells were co-treated with compound **13** at concentrations of 5 μM and 10 μM, IL-6 expression was notably downregulated (Figure 10A,B).

### 3.9. Compound ***13*** Inhibited 4-TBP-Induced Upregulation of Key UPR Proteins

Vitiligo patients exhibit unpolished protein accumulation in melanocytes [10]. When melanocytes undergo continuous intracellular stress, the excessive release of ROS and inflammatory factors induces the activation of the unfolded protein reactions (UPR), ultimately leading to apoptosis [15]. Previous research has shown that 4-TBP stimulates PIG3V melanocytes, activating UPR and causing an increase in the expression of IRE1, ATF6, PERK, and their downstream target, ATF4 and EIF2*α* phosphate [45]. The experimental results demonstrated that compound **13** can inhibit ROS and IL-6 levels in 4-TBP-stimulated PIG3V melanocytes. Therefore, it is necessary to determine whether compound **13** has a regulatory effect on UPR-related proteins. In this section, we determine the protein expression of Bip, IRE1, XBP-1, ATF6, PERK, ATF4, and EIF2*α* using immunoblotting. As shown in Figure 11, the expression of Bip, IRE1, XBP-1, ATF6, PERK, ATF4, and EIF2*α* in the normal groups is very low. Compared to the normal group, the 4-TBP model group exhibited increased expression of Bip, IRE1, XBP-1, ATF6, PERk, ATF4, and EIF2*α*. In comparison to the model group, the compound **13** intervention group showed a significant concentration-dependent decrease in the expression of Bip, IRE1, and XBP-1, with the medium and high dose group of compound **13** displaying significant differences (**** *p* < 0.0001). In contrast, the expression of ATF6, PERk, ATF4, and EIF2*α* remained unchanged in the compound **13** intervention groups compared to the model group.

### 3.10. Compound ***13*** Inhibited 4-TBP-Induced Production of IL-6 by Interfering with UPR

XBP-1 is known to regulate cytokine IL-6 expression. ELISA analysis revealed that within 6 h of exposure to 4-TBP, IL-6 secretion was substantially higher in PIG3V melanocytes than that in controls. Two inhibitors of XBP-1 splicing, namely salicylaldehyde (SA) [46] and rapamycin [47], were employed to impede the IRE1 arm of the UPR in melanocytes both prior to and subsequent to the 4-TBP treatment. We hypothesized that compound **13** could reduce ROS generation and downregulate IL-6 production by interfering with UPR. Therefore, to verify this hypothesis, PIG3V cells treated with compound **13** were exposed to both rapamycin and SA. We found that rapamycin and SA (Figure 12) further augmented compound **13**-induced downregulation in the production of IL-6. XBP-1 is a transcription factor that modulates downstream UPR targets, and its splicing is induced by the activation, dimerization, and phosphorylation of IRE1. Activation of XBP-1 was involved in 4-TBP-induced IL-6 expression that might contribute to the autoimmune-mediated progression of vitiligo. XBP-1 inhibition downregulated IL-6 production, eventually decreasing the load on the folding machinery in ER stress and UPR. These results indicated that compound **13** inhibited 4-TBP-induced production of IL-6 by interfering with UPR.

## 4. Discussion

Vitiligo, a pervasive chronic autoimmune skin disorder, is marked by the obliteration of melanocytes within the skin [48]. The etiology of vitiligo is multifaceted, with autoimmunity, oxidative stress, and environmental factors implicated, signifying a complex interplay of components working in unison to trigger and sustain vitiligo’s progression [49]. Consequently, we surmised that enhancing melanogenesis alone is insufficient as a therapeutic target for vitiligo, necessitating the exploration of other underlying mechanisms associated with the compounds under investigation. PIG3V cells, immortalized human vitiligo melanocytes, exhibit both ER dilatation and inherent defects, rendering them more susceptible to oxidative stress [9]. Persistent stress in melanocytes precipitates apoptosis, subsequently instigating the immune system’s targeting of melanocytes [7].

In traditional Chinese medicine, *Ruta graveolens* L. has been used as a folk medicine to treat vitiligo [24]. To identify its active components and elucidate its anti-vitiligo mechanism, we isolated compounds from the plant extracts and evaluated their structure and activity. Sixteen alkaloids were isolated and identified using nuclear magnetic resonance (NMR), high-resolution mass spectrometry (HRESIMS), ultraviolet spectroscopy (UV), infrared spectroscopy (IR), circular dichroism (CD), and a literature data comparison. A new C-6 hydroxy substituted dihydropyrano [2,3-*b*] quinoline alkaloid rutagrarin A (**1**) was found (**2**–**16**). Among them, compounds (**2**, **3**, **8**, **10**) were isolated from *R. graveolens* for the first time. Upon screening these compounds, we discovered that compound **13** significantly promoted melanogenesis and tyrosinase activity in a concentration-dependent manner in B16 cells.

PIG3V, as immortalized human vitiligo melanocytes characterized by ER dilation and intrinsic defects, is more susceptible to oxidative stress. Sustained melanocyte stress will lead to melanocyte apoptosis. This study sought to identify mechanisms of compound **13** for ER stress and the UPR activating in human melanocytes. We evaluated whether compound **13** could decrease the load on the 4-TBP-induced folding machinery in ER stress and UPR. 4-TBP, a trigger of vitiligo, induces ROS generation, leading to disruption of homeostasis in the ER of melanocytes. The subsequent accumulation of protein folding perturbations activates UPR, which may be a pathway implicated in determining susceptibility to vitiligo. PIG3V melanocyte exposure to 4-TBP has been shown to induce apoptosis [10]. In the present study, at a concentration of 300 µM, 4-TBP leads to a ≥20% loss of cell viability. However, compound **13** -increased the cell survival rate, indicating that it protected PIG3V melanocytes against 4-TBP-induced oxidative injury.

ROS generation in melanocytes is considered a provocative mechanism responsible for cytotoxicity induced by 4-TBP [12]. Evidence supports that 4-TBP can generate ROS in melanocytes, leading to misfolded peptides that subsequently activate the UPR signaling pathway [10]. IRE1 is one of the signaling nodes of UPR. IRE1 is activated through sequential dimerization and trans-autophosphorylation to splice XBP-1, which increases XBP-1 expression and IL-6 production. Increased IL-6 expression may contribute to the autoimmune-mediated progression of vitiligo [50]. Therefore, ER stress-induced UPR also activates innate inflammation.

Our results revealed that IL-6 expression was downregulated in PIG3V cells co-treated with compound **13** and 300 μM of 4-TBP. Western blot analysis indicated that increased expression and phosphorylation of IRE1, Bip, and XBP-1 were downregulated after co-treatment with 4-TBP and compound **13**. In addition, this compound-induced downregulation in the production of IL-6 was further augmented by two XBP-1 inhibitors, rapamycin and SA. Therefore, we believed that compound **13** exerted its therapeutic action by managing ER stress and preventing autoimmune-mediated progression to reduce the worsening of vitiligo.

## 5. Conclusions

In conclusion, we isolated sixteen compounds from the herb of *R. graveolens,* including a previously undescribed quinoline alkaloid named rutagrarin A (**1**). Subsequently, we used B16 cells to study the melanogenic effects of *R. graveolens*. The results showed that the main chemical composition of the chloroform fraction of *R. graveolens* was alkaloids. The extract of *R. graveolens* displayed potent anti-vitiligo activity. Among the isolated constituents, compound **13** exhibited significant anti-vitiligo activities. In addition, this study sought to identify the mechanism of action of the detected compounds in ER stress and UPR activation in human melanocytes. We analyzed the effects of compound **13** on 4-TBP-induced oxidative injury and UPR signals in PIG3V melanocytes. Therefore, we believed that **13** exerted its therapeutic action by managing ER stress and preventing autoimmune-mediated progression to reduce the worsening of vitiligo. Overall, the present study at least partially elucidated the chemical constituents of the chloroform fraction of the herb *R. graveolens* and revealed that the compounds from *R. graveolens* extracts upregulated melanogenesis in melanocytes, managed ER stress, and prevented autoimmune-mediated progression to reduce the worsening of vitiligo, which provide the scientific basis of this plant as the herb in the folk application.

## Figures and Tables

**Figure 1 antioxidants-12-01580-f001:**
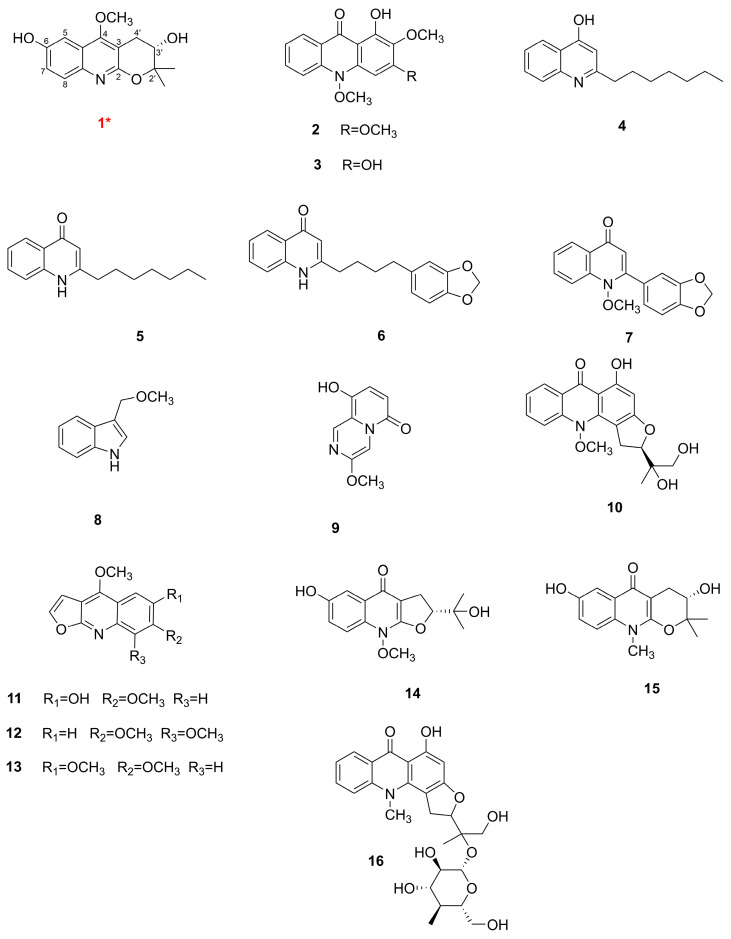
The structures of the compounds isolated from *R. graveolens,* *—new compound.

**Figure 2 antioxidants-12-01580-f002:**
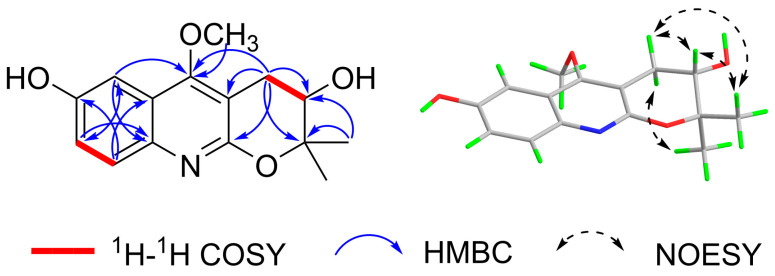
Selected key ^1^H–^1^H COSY, HMBC, and NOESY correlations of compound **1**.

**Figure 3 antioxidants-12-01580-f003:**
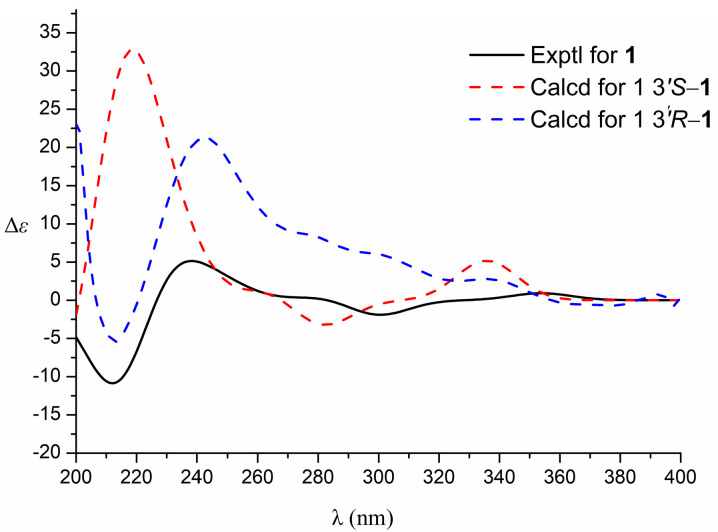
Experimental and calculated ECD spectra of compound **1**.

**Figure 4 antioxidants-12-01580-f004:**
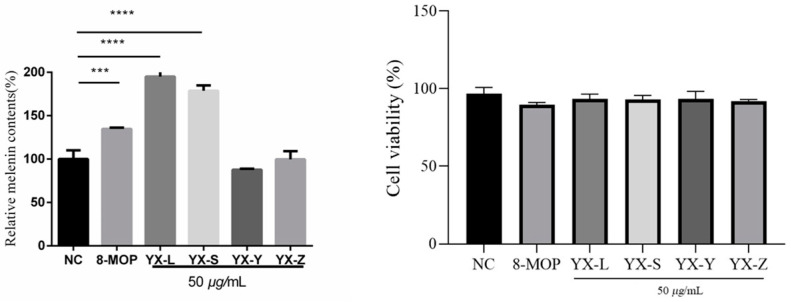
Preliminary screening for anti-vitiligo activities of different polarity fractions from *R. graveolens.* NC represents the solvent DMSO control group; 8-Methoxypsoralen (8-MOP) is a positive group with a concentration of 50 µm compared with the blank control group NC (n = 3; *** *p* < 0.001, **** *p* < 0.0001); ns: there is no statistical difference. YX-L is chloroform extract; YX-S is petroleum ether extract; YX-Y is ethyl acetate extract; YX-Z is *n*-butanol extract.

**Figure 5 antioxidants-12-01580-f005:**
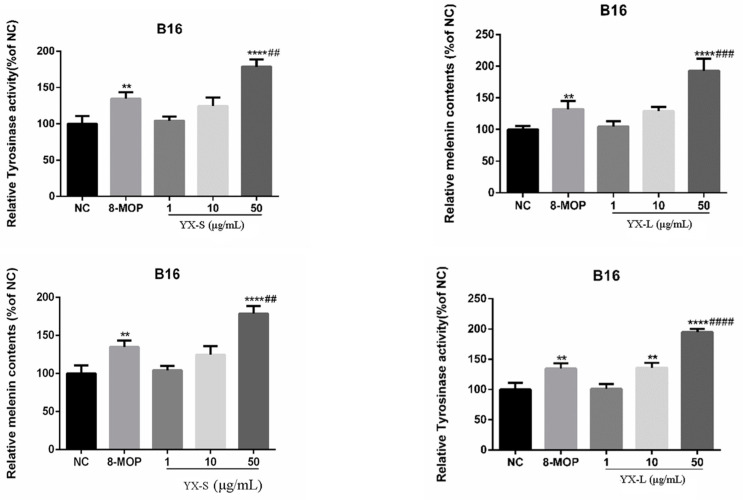
Effect of petroleum ether and chloroform fractions of *R. graveolens* on tyrosinase activity and melanin contents. NC represents the solvent DMSO control group; 8-Methoxypsoralen (8-MOP) is a positive control group with a concentration of 50 µm (n = 3; ** *p* < 0.01, **** *p* < 0.0001 compared with the blank control group NC; ^##^
*p* < 0.01, ^###^ *p* < 0.001, ^####^ *p* < 0.0001, compared with the positive control group).

**Figure 6 antioxidants-12-01580-f006:**
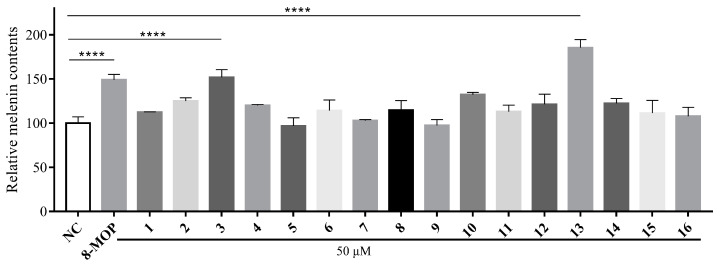
Stimulation of melanin content of B16 cells by chloroform fractions of *R. graveolens.* Results were presented as the mean ± SD (n = 3), **** *p* < 0.0001.

**Figure 7 antioxidants-12-01580-f007:**
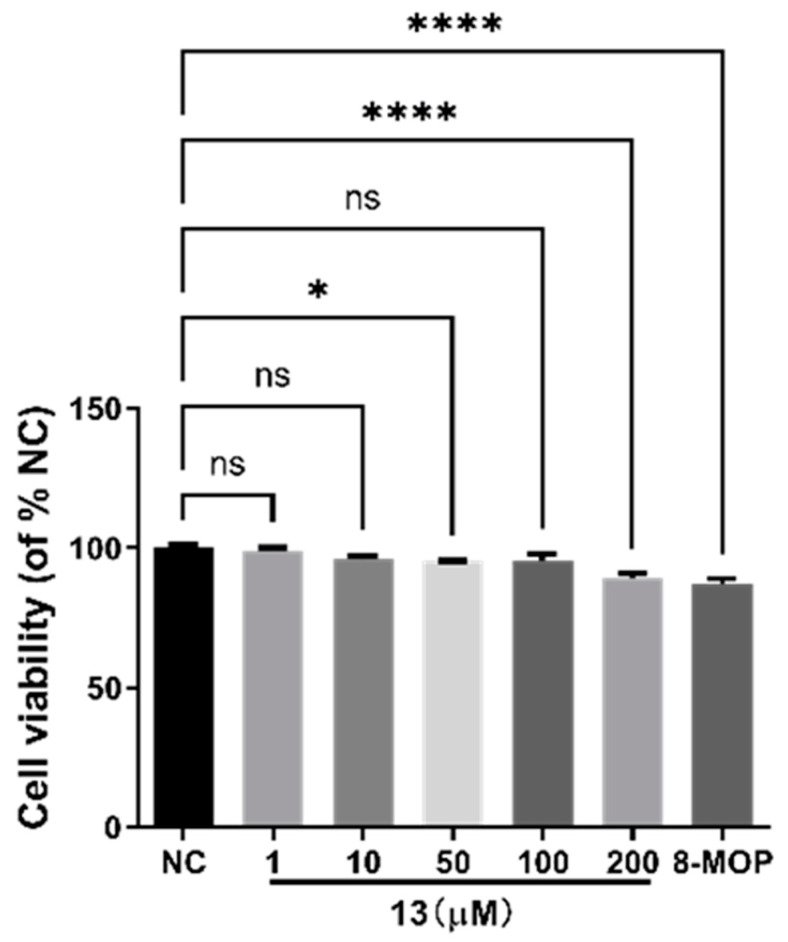
Stimulation of melanin content of B16 cells by compound **13** of *R. graveolens*. NC is negative control; 8-MOP is positive control; * *p* < 0.05, **** *p* < 0.0001 versus untreated control; n = 3. NS: Not statistically significant.

**Figure 8 antioxidants-12-01580-f008:**
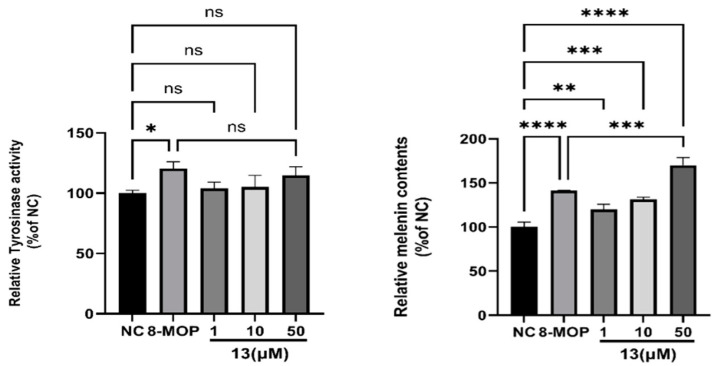
Effect of **13** on melanin contents and TYR activity in B16 melanoma cells. NC is negative control; 8-MOPis positive control; * *p* < 0.05, ** *p* < 0.01, *** *p* < 0.001, **** *p* < 0.0001 versus untreated control; n = 3. NS: Not statistically significant.

**Figure 9 antioxidants-12-01580-f009:**
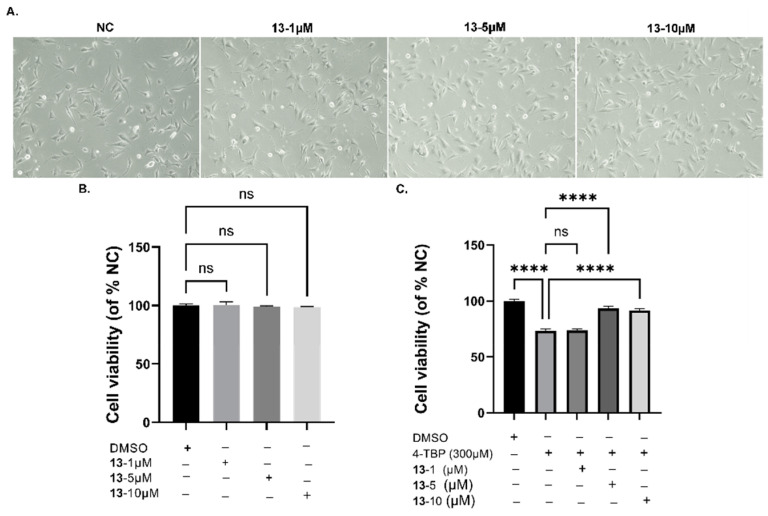
Effects of **13** on 4-TBP-induced oxidative injury in PIG3V melanocytes: (**A**) Morphological changes in compound **13**-treated PIG3V cells at 200× magnification. (**B**) Effects of different concentrations (0–10 µM) of **13** on the cell viability of PIG3V melanocytes. (**C**). Effects of different concentrations (0–10 µM) of compound **13** after 4-TBP-induced oxidative injury measured by the CCK-8 assay (**** *p* < 0.0001 versus 0 h, n = 3). NS: Not statistically significant.

**Figure 10 antioxidants-12-01580-f010:**
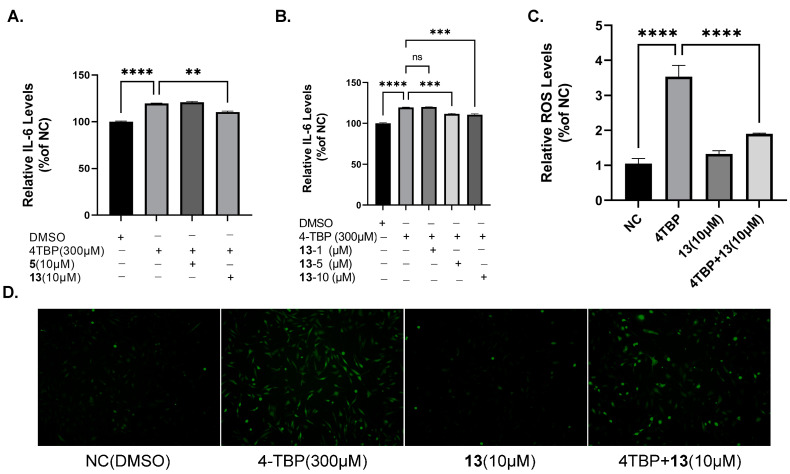
Effects of compound **13** on 4-TBP-induced ROS generation and cytokine IL-6 expression in PIG3V cells: (**A**,**B**) ELISA indicates decreased IL-6 levels in PIG3V melanocytes treated with **13** versus 4-TBP controls (** *p* < 0.01 versus 4-TBP controls, *** *p* < 0.001 versus 4-TBP controls). (**C**) Measurement of fluorescence intensities of the DCFH-DA probe at an excitation of 500 nm and emission of 530 nm after ROS generation (**** *p* < 0.0001 versus 4-TBP controls, n = 3). (**D**) Micrographs taken under a fluorescence microscope. NS: Not statistically significant.

**Figure 11 antioxidants-12-01580-f011:**
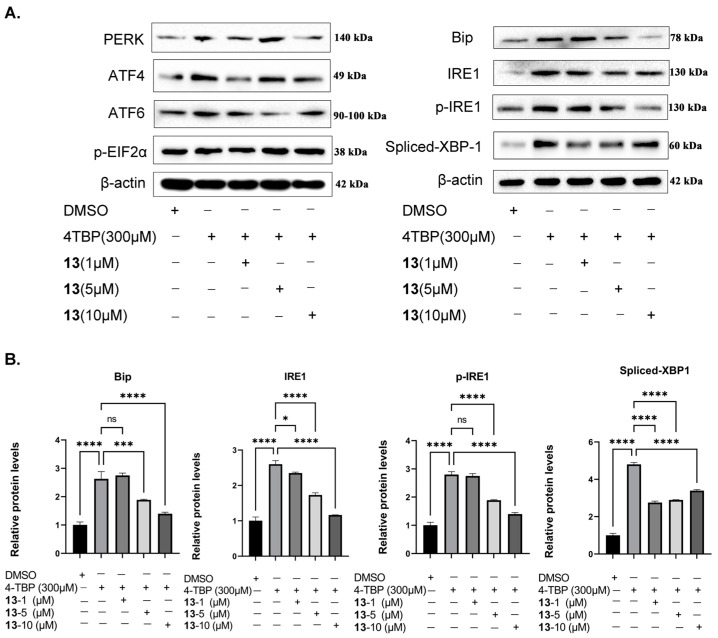
Compound **13** inhibits 4-TBP-induce upregulation of the IRE1 initiation arm of UPR: (**A**) Western blot analysis of Bip, IRE1, p-IRE1, and XBP-1s in PIG3V melanocytes treated with 4-TBP and compound **13**. (**B**) Expression levels of each protein. The protein band density determined using the Photoshop program (* *p* < 0.0001 versus 4-TBP controls, *** *p* < 0.001, **** *p* < 0.0001 versus 4-TBP controls, n = 3). NS: Not statistically significant.

**Figure 12 antioxidants-12-01580-f012:**
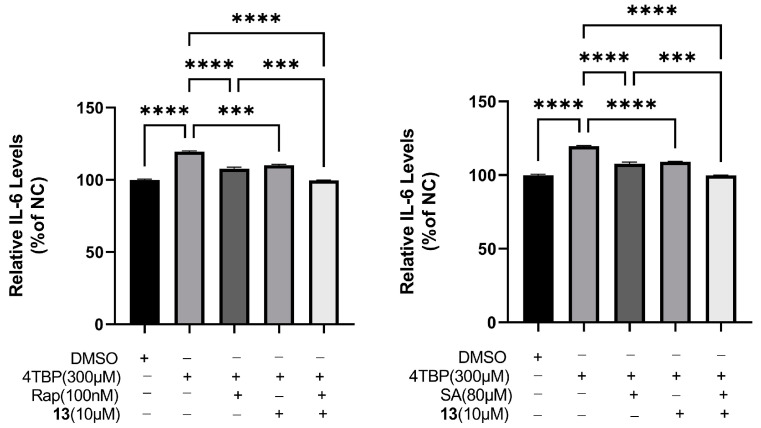
Compound **13** inhibited 4-TBP-induced production of IL-6 by interfering with UPR. Decreased production of IL-6 in **13**-treated PIG3V cells exposed to either rapamycin or salicylaldehyde (SA) after treatment with 4-TBP, and decreased expression levels of IL-6 after co-treatment with **13** versus 4-TBP controls (*** *p* < 0.001, **** *p* < 0.0001, n = 3).

**Table 1 antioxidants-12-01580-t001:** ^1^H and ^13^C NMR spectra data of the compounds **1** (in CD_3_OD).

No.	1	
	^1^H	^13^C
2		159.6
3		109.5
4		164.0
5	7.23 (1H, d, *J* = 2.8 Hz)	104.4
6		155.3
7	7.19 (1H, dd, *J* = 9.0, 2.8 Hz)	122.9
8	7.56 (1H, d, *J* = 9.0 Hz)	128.5
4a		122.7
8a		142.5
2′		80.4
3′	3.90 (1H, dd, *J* = 6.6, 5.1 Hz)	69.2
4′*α*	2.94 (1H, dd, *J* = 17.1, 6.6 Hz)	
4′*β*	3.21 (1H, dd, *J* = 17.1, 5.1 Hz)	
2′*α*-CH_3_	1.41(3H, s)	22.2
2′*β*-CH_3_	1.44(3H, s)	26.1
4-OCH_3_	4.01(3H, s)	61.6

## Data Availability

The data presented in this study are available in the Appendix A.

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
