# Peer review of "Ruta graveolens: Boost Melanogenic Effects and Protection against Oxidative Damage in Melanocytes"

_antioxidants, 2023, doi:10.3390/antiox12081580_

Round 1

Reviewer 1 Report

The author studied melanin production and antioxidant efficacy of Ruta graveolens and its components for vitiligo relief. However, it seems that some errors need to be corrected and additional explanations are needed.

1) How long did the author treat the sample to B16 cells to measure the amount of melanin? Usually, melanin production can be observed well after about 3 days of sample treatment.

2) Did the author measure the amount of melanin secreted outside the cell, not the amount of melanin inside the cell? When melanin production increases in B16 cells, it can be easily observed that the color of the culture media changes to black.

3) Did the author observe cell proliferation about 3 days, not toxicity of B16 cells after component treatment? In addition to melanin increase, melanocyte proliferation also acts as an important mechanism for vitiligo relief.

4) Did the author check whether component 13 affects tyrosinase expression if it does not affect tyrosinase activity in Figure 9?

5) Check the component treatment codes (+/-) in Figure 11B and 13 and correct any errors. The same or incorrect notation is repeated.

6) What do 'SA 50' and 'rapamycin 51' indicated on the 443-444 line on page 14 mean? If referenced, check notation and reference number.

Reviewer 2 Report

The manuscript presents a justification for the traditional usage of Ruta graveolens for vitiligo treatment. The subject is interesting, and the experimental part is well planned. The description of the results is clear; however, some points should be corrected/explained. Generally, the manuscript should be prepared with more care. There are many typographical errors, such as unnecessary capital letters (e.g., see lines 196, 264, in the names of compounds...) or lack of capital letters (line 265, in the title of some sections...). Please check all the text carefully.

The other comments:

Correct the title. It is grammatically incorrect.

There is no information about the preparation of the compound solution for the cell line assay (e.g., what solvent was used?)

2.1. Explain the abbreviation: FBS, PNPP

2.2. “The aerial parts of R. graveolens” – what part exactly was used? Leaves, stems and flower?

Line 97: “Various column chromatography (CC) such as silica gel column..” – correct: “silica gel column chromatography”, add more detail on OSD column

Line 102: add the volume of ethanol

Line 163: add “cells/well”

Line 201: the word: „technique” is unnecessary

Figure 1: Explain in the figure legend why number 1 is in red color.

The structures numbered 4 and 5 are the same.

Figure 2 is unnecessary as the structure is already shown in Figure 1.

For Figure 5 and 6, the legend states that "NC represents the solvent (DMSO) control" – is it pure DMSO? According to the literature, DMSO is highly cytotoxic. Please explain the abbreviation "8-MOP" and the meaning of "####" in the figure legend.

Line 282: „promoted melanin production at 50 μg/ml, reaching 195.1% 282± 5.120 and 178.7 ± 6.239” – In Figure 5, the values are significantly lower. Please add '%' to all the mentioned values

Line 298: „within the concentration range of…” – this is not concentration range: „at concentration of…

Line 315: „This suggests that the chloroform and petroleum  ether extracts may be the key active components responsible for R. graveolens’s ability….” – I think that „ extract may contein key active commponents” should be better. Furthermore, the sentence is too long.

Line 327: „212.7% and 185.3%” – why two values? Figure 7 contains only one panel.

Figure 7: Complete the figure legend. It should be self-explanatory.

Line 336: „within  the concentration range of 1-50 μM, compound 13 has no significant impact on B16 cell” – on Figure 8 the difference for 50 μM is marked as statistically significant

Figure 8, 9: „P means positive control” - no P is shown on figure

Lines 415-419: add appropriate reference

Figure 12b is not visible. The fonts are too small

Reviewer 3 Report

1. What is the primary chemical constituent of Ruta graveolens L., which has been clinically utilized for vitiligo treatment?

2. In vitiligo patients, what abnormality do melanocytes exhibit compared to control cells when exposed to oxidative stress?

3. What are the major functional groups present in compound 1 according to the 13C-NMR and HMQC analyses?

4. How did the authors determine the absolute configuration of compound 1?

5. What is the systematic name of compound 1, and what is its common name?

6. How did the researchers assess the melanogenic effects of R. graveolens fractions in B16 cells, and which fraction showed the most significant melanin promotion?

7. What was the role of XBP-1 in 4-TBP-induced IL-6 expression, and how did XBP-1 inhibition affect IL-6 production in PIG3V cells treated with compound 13 and 4-TBP?

8. What potential therapeutic implications does compound 13 hold for vitiligo treatment, considering its effects on melanin production and UPR interference?

9. How did compound 13 protect PIG3V melanocytes against 4-TBP-induced oxidative injury?

Moderate editing of English language required

Reviewer 4 Report

Overview of the manuscript
The work focuses on the analysis of the protective effect of the alkaloid extract from Ruta graveolens on the melanin production by melanocytes. The authors perform a different extraction protocol, obtaining several alkaloids.

The effect of different extraction groups was verified on a melanocyte cell line to verify the biological effect in promoting melanin production.

In particular, an alkaloid has been studied more carefully, because the preliminary results showing a more evident effect in promoting melanin production. This alkaloid has shown to strongly inhibit the oxidative stress associated to ER oxidative stress pathways in melanocytes, increasing melanin production. The authors propose this alkaloid as a potential therapeutic agent against vitiligo.

GENERAL COMMENT

The topic of the work is very interesting. The authors perform adequate and well analytic procedure to characterise the extract compounds. The use in the biological experiments of PIG3V Melanocytes adds interest to the topic of the work. The experimental plane is well performed, and the methodological approach is suitably constructed and rich, giving adequate support to results and conclusions. However, the manuscript should be improved in its presentation, in particular the Material and Methods section is not adequately presented, and some points result enigmatic and difficult to understand.

 SPECIFIC COMMENTS

Abstract

Avoid the indicative number of compounds, they can be identifiable only after the reading of the work text.

Pag. 1, line 10: “…three major refractory skin diseases…”. Explain which they are, or not mentioned them.

Pag. 1, line 23: your focus is the oxidative stress and the resulting production of inflammatory cytokine, the “autoimmune-mediated progression” is not your issue.

Introduction

In Introduction section, the use of PIG3V melanocytes should be adequately introduced, their use being emphasized in the title.

Pag. 1, line 40-41: what does “white spot” means. Rephrased the sentence.

Pag. 2, line 60-62: add references to the sentence.

Materials and Methods

The term “sample” along the section remains confusing. Do you indicate the complex of polarity fractions extract? Use different term and explain better.

Pag. 3, line 133: what is CC7. Explain.

Pag. 3, line 141: “cultured pigment cells”? Which type of cells have you used. Explain better.

Pag. 4, line 148: “incubated with the component” which component? Explain better.

Pag. 4, line 152-161: which primary antibodies have you used, give the specifics in this paragraph also.

Pag. 4, paragraph 2.8 and 2.9: the use of different concentration scales remain not clear. Explain better.

Pag. 4, line 172: which concentration for 4-TBP? The same as above. Explain.

Results

Pag. 6, line 208: Is compound 1 the new characterised compound? Be explicit.

Pag. 8, line 283: What is the “8-MOP positive control”? Its use is not stated in Mat. and Met. section. In what respect it represents a control. Explain.

Pag. 8, line 305-306: the sentence is not clear. You declare 10-50 μg/ml the optimal concentration. But this range is the one you have effectively used. Your conclusion is an assumption or a result. Explain better.

Pag. 11, line 372-375: this explanatory paragraph should be moved in the Intro section.

Pag. 12, Fig. 11: bar graphs are not easy to understand. In particular in graph A, are the compounds 5 and 13 compared? In graph B, is the compound 13 contemporary administrated in several concentration on the same sample? Be clearer.

Pag. 14, line 443: the sentence is not clear, what are the control cells of PIG3V in this experiment? Explain better.

Pag. 14, line 443-444: salicylaldehyde50 and rapamycin51 have not been indicated in Mat and Met section. In particular, this experiment is not indicated in Mat and Met section. Provide to do it.

Discussion

In Discussion section the identification of the new compound should be commented.

minor editing for the syntax

Round 2

Reviewer 1 Report

Acceptable.

Reviewer 3 Report

No comments

Reviewer 4 Report

All observations have been included in the new manuscript presentation

No more concerns